# ANYTEXT: MULTILINGUAL VISUAL TEXT GENERATION AND EDITING

**Yuxiang Tuo, Wangmeng Xiang, Jun-Yan He, Yifeng Geng\*, Xuansong Xie**
Institute for Intelligent Computing, Alibaba Group
{yuxiang.tyx,wangmeng.xwm,leyuan.hjy,cangyu.gyf,xingtong.xxs}
@alibaba-inc.com

## ABSTRACT

Diffusion model based Text-to-Image has achieved impressive achievements recently. Although current technology for synthesizing images is highly advanced and capable of generating images with high fidelity, it is still possible to give the show away when focusing on the text area in the generated image, as synthesized text often contains blurred, unreadable, or incorrect characters, making visual text generation one of the most challenging issues in this field. To address this issue, we introduce **AnyText**, a diffusion-based multilingual visual text generation and editing model, that focuses on rendering accurate and coherent text in the image. AnyText comprises a diffusion pipeline with two primary elements: an auxiliary latent module and a text embedding module. The former uses inputs like text glyph, position, and masked image to generate latent features for text generation or editing. The latter employs an OCR model for encoding stroke data as embeddings, which blend with image caption embeddings from the tokenizer to generate texts that seamlessly integrate with the background. We employed text-control diffusion loss and text perceptual loss for training to further enhance writing accuracy. AnyText can write characters in multiple languages, to the best of our knowledge, this is the first work to address multilingual visual text generation. It is worth mentioning that AnyText can be plugged into existing diffusion models from the community for rendering or editing text accurately. After conducting extensive evaluation experiments, our method has outperformed all other approaches by a significant margin. Additionally, we contribute the first large-scale multilingual text images dataset, **AnyWord-3M**, containing 3 million image-text pairs with OCR annotations in multiple languages. Based on AnyWord-3M dataset, we propose AnyText-benchmark for the evaluation of visual text generation accuracy and quality. Our project will be open-sourced soon on `https://github.com/tyxsspa/AnyText` to improve and promote the development of text generation technology.

## 1 INTRODUCTION

Diffusion-based generative models Saharia et al. (2022); Ramesh et al. (2022); Rombach et al. (2022); Zhang & Agrawala (2023) demonstrated exceptional outcomes with unparalleled fidelity, adaptability, and versatility. Open-sourced image generation models (for example, Stable Diffusion Rombach et al. (2022), DeepFloyd-IF DeepFloyd-Lab (2023)), along with commercial services (Midjourney Inc. (2022), DALL-E 2 Ramesh et al. (2022), etc.) have made substantial impacts in various sectors including photography, digital arts, gaming, advertising, and film production. Despite substantial progress in image quality of generative diffusion models, most current open-sourced models and commercial services struggle to produce well-formed, legible, and readable visual text. Consequently, this reduces their overall utility and hampers potential applications.

The subpar performance of present open-source diffusion-based models can be attributed to a number of factors. Firstly, there is a deficiency in large-scale image-text paired data which includes comprehensive annotations for textual content. Existing datasets for large-scale image diffusion model training, like LAION-5B Schuhmann et al. (2022), lack manual annotations or OCR results

---

\*corresponding author

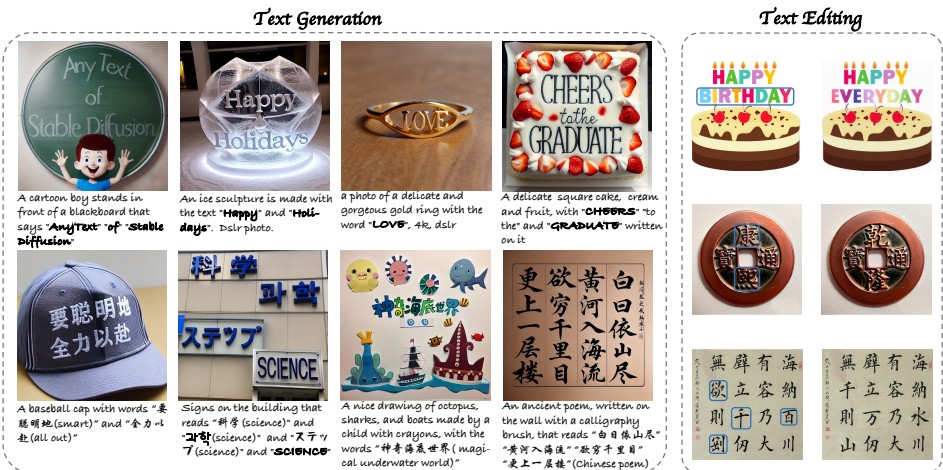

Figure 1: Selected samples of AnyText. For text generation, AnyText can render the specified text from the prompt onto the designated position, and generate visually appealing images. As for text editing, AnyText can modify the text content at the specified position within the input image while maintaining consistency with the surrounding text style. Translations are provided in parentheses for non-English words in prompt, blue boxes indicate positions for text editing. See more in A.6.

Table 1: Comparison of AnyText with other competitors based on functionality.

| Functionality | Multi-line | Deformed regions | Multi-lingual | Text editing | Plug-and-play |
|---|---|---|---|---|---|
| GlyphDraw | ✗ | ✗ | ✗ | ✗ | ✗ |
| TextDiffuser | ✓ | ✗ | ✗ | ✓ | ✗ |
| GlyphControl | ✓ | ✗ | ✗ | ✗ | ✓ |
| AnyText | ✓ | ✓ | ✓ | ✓ | ✓ |

for text content. Secondly, as pointed out in Liu et al. (2023), the text encoder used in open-source diffusion models, such as the CLIP text encoder, employs a vocabulary-based tokenizer that has no direct access to the characters, resulting in a diminished sensitivity to individual characters. Lastly, most diffusion models' loss functions are designed to enhance the overall image generation quality and lack of dedicated supervision for the text region.

To address the aforementioned difficulties, we present *AnyText* framework and *AnyWord-3M* dataset. AnyText consists of a text-control diffusion pipeline with two components: auxiliary latent module encodes auxiliary information such as text glyph, position, and masked image into latent space to assist text generation and editing; text embedding module employs an OCR model to encode stroke information as embeddings, which is then fused with image caption embeddings from the tokenizer to render texts blended with the background seamlessly; and finally, a text perceptual loss in image space is introduced to further enhance writing accuracy.

Regarding the functionality, there are five differentiating factors that set apart us from other competitors as outlined in Table 1: a) *Multi-line*: AnyText can generate text on multiple lines at user-specified positions. b) *Deformed regions*: it enables writing in horizontally, vertically, and even curved or irregular regions. c) *Multi-lingual*: our method can generate text in various languages such as Chinese, English, Japanese, Korean, etc. d) *Text editing*: which provides the capability for modifying the text content within the provided image in consistent font style. e) *Plug-and-play*: AnyText can be seamlessly integrated with stable diffusion models and empowering them with the ability to generate text. We present some selected examples in Fig. 1 and Appendix A.6.

## 2 RELATED WORKS

**Text-to-Image Synthesis**. In recent years, significant strides has been made in text-to-image synthesis using denoising diffusion probabilistic models Ho et al. (2020); Ramesh et al. (2021); Song et al. (2021); Dhariwal & Nichol (2021); Nichol & Dhariwal (2021); Saharia et al. (2022); Ramesh et al. (2022); Rombach et al. (2022); Chang et al. (2023). These models have advanced beyond simple image generation and have led to developments in interactive image editing Meng et al. (2022); Gal et al. (2023); Brooks et al. (2022) and techniques incorporating additional conditions, such as

masks and depth maps Rombach et al. (2022). Research is also exploring the area of multi-condition controllable synthesis Zhang & Agrawala (2023); Mou et al. (2023); Huang et al. (2023). Compositing subjects into scenes presents more specific challenges, and approaches like ELITE Wei et al. (2023), UMM-Diffusion Ma et al. (2023b), and InstantBooth Shi et al. (2023) utilize the features from CLIP image encoder to encode the visual concept into textual word embeddings. Similarly, DreamIdentity Chen et al. (2023c) developed a specially designed image encoder to achieve better performance for the word embedding enhancing scheme.

**Text Generation**. Progress in image synthesis has been substantial, but integrating legible text into images remains challenging Rombach et al. (2022); Saharia et al. (2022). Recent research has focused on three key aspects of text generation:

*Control Condition.* Introducing glyph condition in latent space has been a predominant approach in many recent methods. GlyphDraw Ma et al. (2023a) originally used an explicit glyph image as condition, with characters rendered at the center. GlyphControl Yang et al. (2023) further extends it by aligning the text based on its location, which also incorporates font size and text box position in an implicit manner. TextDiffuser Chen et al. (2023b) utilizes a character-level segmentation mask as control condition, and in addition, it introduces a masked image to simultaneously learn text generation and text in-painting branches. In our work, we adopt a similar way as GlyphControl to render glyph image but incorporate position and masked image as additional conditions. This design enables AnyText to generate text in curved or irregular regions, and handle text generation and editing simultaneously.

*Text Encoder.* The text encoder plays a crucial role in generating accurate visual text. Recent methods such as Imagen Saharia et al. (2022), eDiff-I Balaji et al. (2022), and Deepfloyd IF DeepFloyd-Lab (2023) achieve impressive results by leveraging large-scale language models (e.g., T5-XXL). However, most image generation models still rely on character-blind text encoders, and even character-aware text encoders struggle with non-Latin text generation like Chinese, Japanese, and Korean Liu et al. (2023). To address Chinese rendering, GlyphDraw fine-tunes the text encoder on Chinese images and utilizes the CLIP image encoder for glyph embeddings Ma et al. (2023a). DiffUTE replaces the text encoder with a pre-trained image encoder to extract glyphs in image editing scenarios Chen et al. (2023a). In AnyText, we propose a novel approach to transform the text encoder by integrating semantic and glyph information. This aims to achieve seamless integration of generated text with the background and enable multi-language text generation.

*Perceptual Supervision.* OCR-VQGAN Rodriguez et al. (2023) employs a pre-trained OCR detection model to extract features from images, and supervise text generation by constraining the differences between multiple intermediate layers. In contrast, TextDiffuser Chen et al. (2023b) utilizes a character-level segmentation model to supervise the accuracy of each generated character in the latent space. This approach requires a separately trained segmentation model, and the character classes are also limited. In AnyText, we utilize an OCR recognition model that excels in stroke and spelling to supervise the text generation within the designated text region only. This approach provides a more direct and effective form of supervision for ensuring accurate and high-quality text generation.

## 3 METHODOLOGY

As depicted in Fig. 2, the AnyText framework comprises a text-control diffusion pipeline with two primary components (auxiliary latent module and text embedding module). The overall training objective is defined as:

$$\mathcal{L} = \mathcal{L}_{td} + \lambda * \mathcal{L}_{tp} \tag{1}$$

where $\mathcal{L}_{td}$ and $\mathcal{L}_{tp}$ are text-control diffusion loss and text perceptual loss, and $\lambda$ is used to adjust the weight ratio between two loss functions. In the following sections, we will introduce the text-control diffusion pipeline, auxiliary latent module, text embedding module, and text perceptual loss in detail.

### 3.1 TEXT-CONTROL DIFFUSION PIPELINE

In the text-control diffusion pipeline, we generate the latent representation $z_0 \in \mathbb{R}^{h \times w \times c}$ by applying Variational Autoencoder (VAE) Kingma & Welling (2014) $\mathcal{E}$ on the input image $x_0 \in \mathbb{R}^{H \times W \times 3}$.

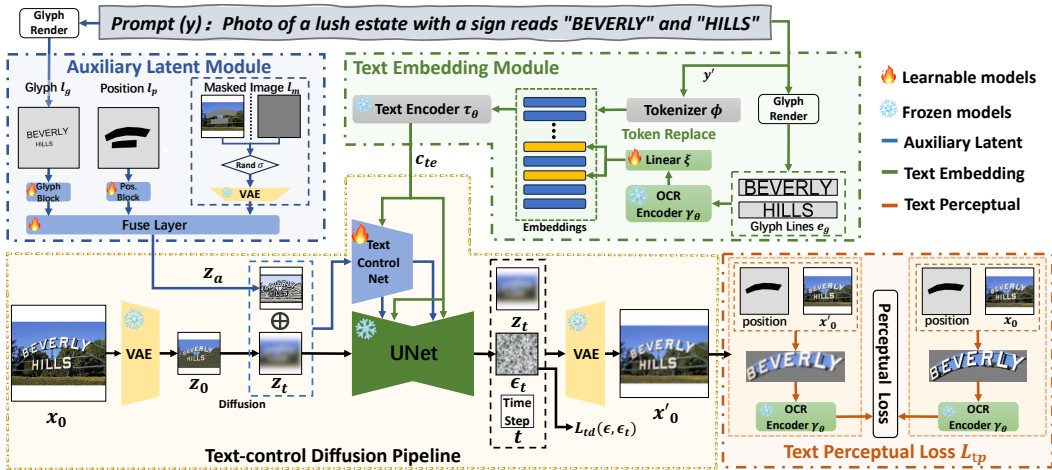

Figure 2: The framework of AnyText, which includes text-control diffusion pipeline, auxiliary latent module, text embedding module, and text perceptual loss.

Here, $h \times w$ represents the feature resolution downsampled by a factor of $f$, and $c$ denotes the latent feature dimension. Then latent diffusion algorithms progressively add noise to the $z_0$ and produce a noisy latent image $z_t$, where $t$ represents the time step. Given a set of conditions including time step $t$, auxiliary feature $z_a \in \mathbb{R}^{h \times w \times c}$ produced by auxiliary latent module, as well as text embedding $c_{te}$ produced by text embedding module, text-control diffusion algorithm applies a network $\epsilon_\theta$ to predict the noise added to the noisy latent image $z_t$ with objective:

$$\mathcal{L}_{td} = \mathbb{E}_{z_0, z_a, c_{te}, t, \epsilon \sim \mathcal{N}(0,1)} \left[ \|\epsilon - \epsilon_\theta(z_t, z_a, c_{te}, t)\|_2^2 \right] \tag{2}$$

where $\mathcal{L}_{td}$ is the text-control diffusion loss. More specifically, to control the generation of text, we add $z_a$ with $z_t$ and fed them into a trainable copy of UNet's encoding layers referred to as TextControlNet, and fed $z_t$ into a parameter-frozen UNet. This enables TextControlNet to focus on text generation while preserving the base model's ability to generate images without text. Moreover, through modular binding, a wide range of base models can also generate text. Please refer to Appendix A.1 for more details.

## 3.2 AUXILIARY LATENT MODULE

In AnyText, three types of auxiliary conditions are utilized to produce latent feature map $z_a$: glyph $l_g$, position $l_p$, and masked image $l_m$. Glyph $l_g$ is generated by rendering texts using a uniform font style (i.e., "Arial Unicode") onto an image based on their locations. Accurately rendering characters in curved or irregular regions is considerably challenging. Therefore, we simplify the process by rendering characters based on the enclosing rectangle of the text position. By incorporating specialized position $l_p$, we can still generate text in non-rectangular regions, as illustrated in Fig. 3. Position $l_p$ is generated by marking text positions on an image. In the training phase, the text positions are obtained either from OCR detection or through manual annotation. In the inference phase, $l_p$ is obtained from the user's input, where they specify the desired regions for text generation. Moreover, the position information allows the text perceptual loss to precisely target the text area. Details regarding this will be discussed in Sec. 3.4. The last auxiliary information is masked image $l_m$, which indicates what area in image should be preserved during the diffusion process. In the text-to-image mode, $l_m$ is set to be fully masked, whereas in the text editing mode, $l_m$ is set to mask the text regions. During training, the text editing mode ratio is randomly switched with a probability of $\sigma$.

To incorporate the image-based conditions, we use glyph block and position block to downsample glyph $l_g$ and position $l_p$, and VAE encoder $\mathcal{E}$ to downsample the masked image $l_m$, respectively. The glyph block $G$ and position block $P$ both contain several stacked convolutional layers. After transforming these image-based conditions into feature maps that match the spatial size of $z_t$, we utilize a convolutional fusion layer $f$ to merge $l_g$, $l_p$, and $l_m$, resulting in a generated feature map denoted as $z_a$, which can be represented as:

$$z_a = f(G(l_g) + P(l_p) + \varepsilon(l_m)) \tag{3}$$

where $z_a$ shares the same number of channels as $z_t$.

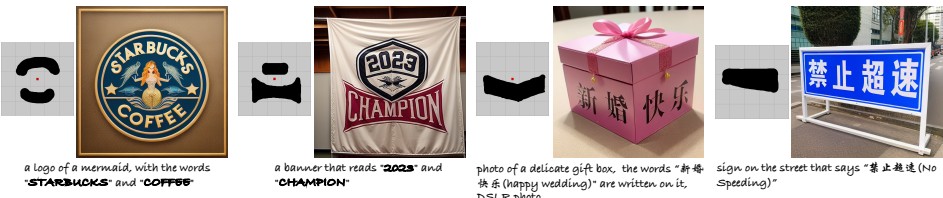

Figure 3: Illustration of generating text in curved or irregular regions, left images are text positions provided by the user.

### 3.3 TEXT EMBEDDING MODULE

Text encoders excel at extracting semantic information from caption, but semantic information of the text to be rendered is negligible. Additionally, most pre-trained text encoders are trained on Latin-based data and can't understand other languages well. In AnyText, we propose a novel approach to address the problem of multilingual text generation. Specifically, we render glyph lines into images, encode glyph information, and replace their embeddings from caption tokens. The text embeddings are not learned character by character, but rather utilize a pre-trained visual model, specifically the recognition model of PP-OCRv3 Li et al. (2022). The replaced embeddings are then fed into a transformer-based text encoder as tokens to get fused intermediate representation, which will then be mapped to the intermediate layers of the UNet using a cross-attention mechanism. Due to the utilization of image rendering for text instead of relying solely on language-specific text encoders, our approach significantly enhances the generation of multilingual text, as depicted in Fig. 4.

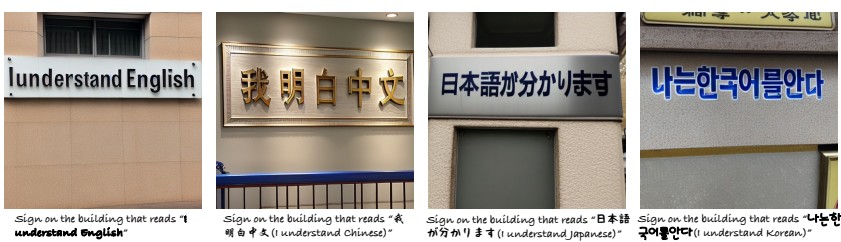

Figure 4: Illustration of generating text in multiple languages.

Next, we provide a detailed explanation of the text embedding module. The representation $c_{te}$ that combines both text glyph and caption semantic information is defined as

$$c_{te} = \tau_\theta(\phi(y'), \xi(\gamma_\theta(e_g)))$$

(4)

where $y'$ is the processed input caption $y$ that each text line to be generated (enclosed within double quotation marks) is replaced with a special placeholder $S_*$. Then after tokenization and embedding lookup denoted as $\phi(\cdot)$, the caption embeddings are obtained. Then, each text line is rendered onto an image, denoted as $e_g$. Note that $e_g$ is generated by only rendering a single text line onto an image in the center, while $l_g$ in Sec. 3.2 is produced by rendering all text lines onto a single image on their locations. The image $e_g$ is then fed into an OCR recognition model $\gamma_\theta$ to extract the feature before the last fully connected layer as text embedding, then a linear transformation $\xi$ is applied to ensure its size matches caption embeddings, and replace it with embedding of $S_*$. Finally, all token embeddings are encoded using a CLIP text encoder $\tau_\theta$.

### 3.4 TEXT PERCEPTUAL LOSS

We propose a text perceptual loss to further improve the accuracy of text generation. Assuming $\varepsilon_t$ represents the noise predicted by the denoiser network $\epsilon_\theta$, we can combine the time step $t$ and noisy latent image $z_t$ to predict $z_0$ as described in Ho et al. (2020). This can be further used with the VAE decoder to obtain an approximate reconstruction of the original input image, denoted as $x'_0$. By transitioning from latent space to image space, we can further supervise text generation at a pixel-wise level. With the help of the position condition $l_p$, we can accurately locate the region of generated text. We aim to compare this region with the corresponding area in the original image $x_0$, and focus solely on the writing correctness of the text itself, excluding factors such as background, deviations in character positions, colors, or font styles. Thus, we employ the PP-OCRv3 model, as mentioned

in Sec. 3.3, as the image encoder. By processing $x_0$ and $x'_0$ at position $p$ through operations such as cropping, affine transformation, padding, and normalization, we obtain the images $p_g$ and $p'_g$ to be used as inputs for the OCR model. We utilize the feature maps $\hat{m}_p, \hat{m}'_p \in \mathbb{R}^{h \times w \times c}$ before the fully connected layer to represent the textual writing information in the original and predicted image at position $p$, respectively. The text perceptual loss is expressed as

$$\mathcal{L}_{tp} = \sum_p \frac{\varphi(t)}{hw} \sum_{h,w} \|\hat{m}_p - \hat{m}'_p\|_2^2 \tag{5}$$

By imposing a Mean Squared Error (MSE) penalty, we attempt to minimize the discrepancies between the predicted and original image in all text regions. As time step $t$ is related to the text quality in predicted image $x'_0$, we need to design a weight adjustment function $\varphi(t)$. It has been found that setting $\varphi(t) = \bar{\alpha}_t$ is a good choice, where $\bar{\alpha}_t$ is the coefficient of diffusion process introduced in Ho et al. (2020).

## 4 DATASET AND BENCHMARK

Currently, there is a lack of publicly available datasets specifically tailored for text generation tasks, especially those involving non-Latin languages. Therefore, we propose *AnyWord-3M*, a large-scale multilingual dataset from publicly available images. The sources of these images include Noah-Wukong Gu et al. (2022), LAION-400M Schuhmann et al. (2021), as well as datasets used for OCR recognition tasks such as ArT, COCO-Text, RCTW, LSVT, MLT, MTWI, ReCTS. These images cover a diverse range of scenes that contain text, including street views, book covers, advertisements, posters, movie frames, etc. Except for the OCR datasets, where the annotated information is used directly, all other images are processed using the PP-OCRv3 Li et al. (2022) detection and recognition models. Then, captions are regenerated using BLIP-2 Li et al. (2023). Please refer to Appendix A.3 for more details about dataset preparation.

Through strict filtering rules and meticulous post-processing, we obtained a total of 3,034,486 images, with over 9 million lines of text and more than 20 million characters or Latin words. We randomly extracted 1000 images from both Wukong and LAION subsets to create the evaluation set called AnyText-benchmark. These two evaluation sets are specifically used to evaluate the accuracy and quality of Chinese and English generation, respectively. The remaining images are used as the training set called AnyWord-3M, among which approximately 1.6 million are in Chinese, 1.39 million are in English, and 10k images in other languages, including Japanese, Korean, Arabic, Bangla, and Hindi. For a detailed statistical analysis and randomly selected example images, please refer to Appendix A.4.

For the AnyText-benchmark, we utilized three evaluation metrics to assess the accuracy and quality of text generation. Firstly, we employed the Sentence Accuracy (Sen. Acc) metric, where each generated text line was cropped according to the specified position and fed into the OCR model to obtain predicted results. Only when the predicted text completely matched the ground truth was it considered correct. Additionally, we employed another less stringent metric, the Normalized Edit Distance (NED) to measure the similarity between two strings. As we used PP-OCRv3 for feature extraction during training, to ensure fairness, we chose another open-source model called DuGuangOCR ModelScope (2023) for evaluation, which also performs excellently in both Chinese and English recognition. However, relying solely on OCR cannot fully capture the image quality. Therefore, we introduced the Frechet Inception Distance (FID) to assess the distribution discrepancy between generated images and real-world images.

## 5 EXPERIMENTS

### 5.1 IMPLEMENTATION DETAILS

Our training framework is implemented based on ControlNet[1], and the model's weights are initialized from SD1.5[2]. In comparison to ControlNet, we only increased the parameter size by 0.34% and the inference time by 1.04%, refer to A.2 for more details. Our model was trained on the AnyWord-3M dataset for 10 epochs using 8 Tesla A100 GPUs. We employed a progressive fine-tuning strategy, where the editing branch was turned off for the first 5 epochs, then activated with a

---

[1]https://github.com/lllyasviel/ControlNet
[2]https://huggingface.co/runwayml/stable-diffusion-v1-5

Table 2: Quantitative comparison of AnyText and competing methods. †is trained on LAION-Glyph-10M, and ‡is fine-tuned on TextCaps-5k. All competing methods are evaluated using their officially released models.

| Methods | English | | | Chinese | | |
|---|---|---|---|---|---|---|
| | Sen. ACC↑ | NED↑ | FID↓ | Sen. ACC↑ | NED↑ | FID↓ |
| ControlNet | 0.5837 | 0.8015 | 45.41 | 0.3620 | 0.6227 | 41.86 |
| TextDiffuser | 0.5921 | 0.7951 | 41.31 | 0.0605 | 0.1262 | 53.37 |
| GlyphControl† | 0.3710 | 0.6680 | 37.84 | 0.0327 | 0.0845 | 34.36 |
| GlyphControl‡ | 0.5262 | 0.7529 | 43.10 | 0.0454 | 0.1017 | 49.51 |
| AnyText-**v1.0** | 0.6588 | 0.8568 | 35.87 | 0.6634 | 0.8264 | **28.46** |
| AnyText-**v1.1** | **0.7239** | **0.8760** | **33.54** | **0.6923** | **0.8396** | 31.58 |

probability of $\sigma = 0.5$ for the next 3 epochs. In the last 2 epochs, we enabled the perceptual loss with a weight coefficient of $\lambda = 0.01$. Image dimensions of $l_g$ and $l_p$ are set to be 1024x1024 and 512x512, while $e_g$, $p_g$, and $p'_g$ are all set to be 80x512. We use AdamW optimizer with a learning rate of 2e-5 and a batch size of 48. During sampling process, based on the statistical information from A.4, a maximum of 5 text lines from each image and 20 characters from each text line were chosen to render onto the image, as this setting can cover the majority of cases in the dataset.

Recently, we found that the quality of OCR annotations in the training data has a significant impact on the text generation metrics. We made minor improvements to the method and dataset, then fine-tuned the original model, resulting in the AnyText-v1.1, which achieved significant performance improvements. Details of the updated model are provided in Appendix A.7. It should be noted that all example images used in this article are still generated by the v1.0 model.

## 5.2 COMPARISON RESULTS

### 5.2.1 QUANTITATIVE RESULTS

We evaluated existing competing methods, including ControlNet Zhang & Agrawala (2023), TextDiffuser Chen et al. (2023b), and GlyphControl Yang et al. (2023), using the benchmark and metrics mentioned in Sec. 4. To ensure fair evaluation, all methods employed the DDIM sampler with 20 steps of sampling, a CFG-scale of 9, a fixed random seed of 100, a batch size of 4, and the same positive and negative prompt words. The quantitative comparison results can be found in Table 2. Additionally, we provide some generated images from AnyText-benchmark in Appendix A.5.

From the results, we can observe that AnyText outperforms competing methods in both Chinese and English text generation by a large margin, in terms of OCR accuracy (Sen.ACC, NED) and realism (FID). It is worth mentioning that our training set only consists of 1.39 million English data, while TextDiffuser and GlyphControl are trained on 10 million pure English data. An interesting phenomenon is that ControlNet(with canny control) tends to randomly generate pseudo-text in the background, evaluation methods as in Chen et al. (2023b) that utilize OCR detection and recognition models may yield low evaluation scores. However, in our metric, we focus only on the specified text generation areas, and we find that ControlNet performs well in these areas. Nevertheless, the style of the generated text from ControlNet appears rigid and monotonous, as if it was pasted onto the background, resulting in a poor FID score. Regarding Chinese text generation, both TextDiffuser and GlyphControl can only generate some Latin characters, punctuation marks or numbers within a Chinese text. In AnyText, our Chinese text generation accuracy surpasses all other methods. In the stringent evaluation metric of Sen. ACC, we achieve an accuracy of over 66%. Additionally, AnyText obtains the lowest FID score, indicating superior realism in the generated text.

### 5.2.2 QUALITATIVE RESULTS

Regarding the generation of the English text, we compared our model with state-of-the-art models or APIs in the field of text-to-image generation, such as SD-XL1.0, Bing Image Creator[3], DALL-E2, and DeepFloyd IF, as shown in Fig. 5. These models have shown significant improvements in text generation compared to previous works . However, there is still a considerable gap between them and professional text generation models. Regarding the generation of Chinese text, the complexity of strokes and the vast number of character categories pose significant challenges. GlyphDraw Ma et al. (2023a) is the first method that addresses this task. Due to the lack of open-source models or APIs, we were unable to conduct quantitative evaluation and instead relied on qualitative comparisons using examples from GlyphDraw paper. Meanwhile, since ControlNet can also generate Chinese text effectively, we included it as well. As shown in Fig. 6, AnyText shows superior integration of

---

[3]https://www.bing.com/create

generated text with background, such as text carved into stone, reflections on signboards with words, chalk-style text on a blackboard, and slightly distorted text influenced by clothing folds.

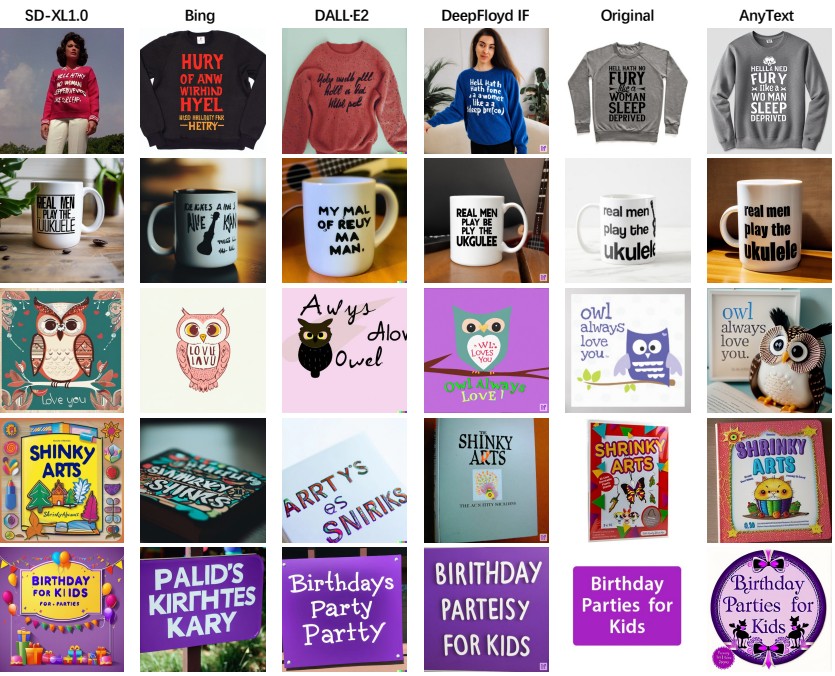

Figure 5: Qualitative comparison of AnyText and state-of-the-art models or APIs in English text generation. All captions are selected from the English evaluation dataset in AnyText-benchmark.

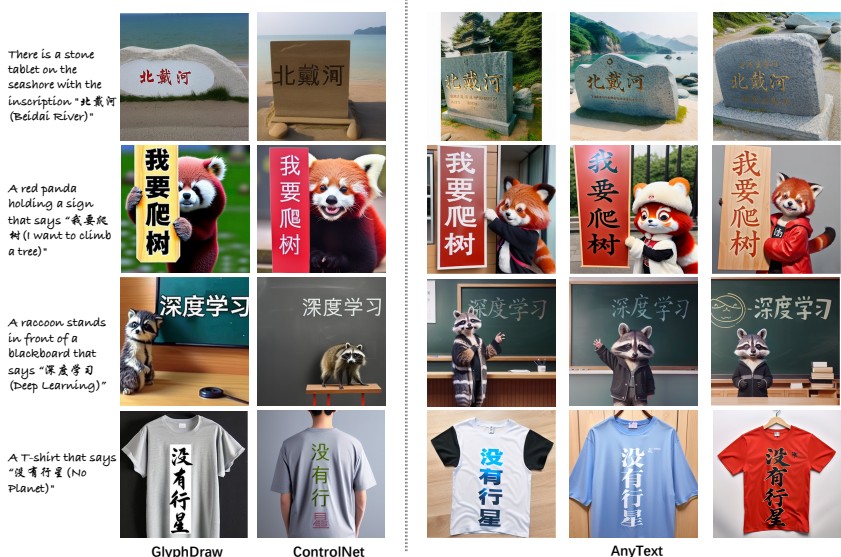

Figure 6: Comparative examples between GlyphDraw, ControlNet, and AnyText in Chinese text generation, all taken from the original GlyphDraw paper.

## 5.3 ABLATION STUDY

In this part, we extracted 200k images (with 160k in Chinese) from AnyWord-3M as the training set, and used the Chinese evaluation dataset in AnyText-benchmark to validate the effectiveness of each submodule in AnyText. Each model was trained for 15 epochs, requiring 60 hours on 8 Tesla V100 GPUs. The training parameters were kept consistent with those mentioned in Sec. 5.1. By analyzing the experimental results in Table 3, we draw the following conclusions:

Table 3: Ablation experiments of AnyText on a small-scale dataset from AnyWord-3M. The results validate the effectiveness of each submodule in AnyText.

| Exp. № | Editing | Position | Text Embedding | Perceptual Loss | $\lambda$ | Chinese | |
|---|---|---|---|---|---|---|---|
| | | | | | | Sen. ACC↑ | NED↑ |
| 1 | ✓ | ✓ | × | × | - | 0.1552 | 0.4070 |
| 2 | × | ✓ | × | × | - | 0.2024 | 0.4649 |
| 3 | × | ✓ | vit | × | - | 0.1416 | 0.3809 |
| 4 | × | ✓ | conv | × | - | 0.1864 | 0.4402 |
| 5 | × | ✓ | ocr | × | - | 0.4595 | 0.7072 |
| 6 | × | × | ocr | × | - | 0.4472 | 0.6974 |
| 7 | × | ✓ | ocr | ✓ | 0.003 | 0.4848 | 0.7353 |
| 8 | × | ✓ | ocr | ✓ | 0.01 | **0.4996** | **0.7457** |
| 9 | × | ✓ | ocr | ✓ | 0.03 | 0.4659 | 0.7286 |

*Editing*: Comparing Exp.1 and Exp.2 we observed a slight decrease when editing branch is enabled. This is because text generation and editing are two different tasks, and enabling editing branch increases the difficulty of model convergence. To focus on analyzing the text generation part, we disabled editing in all subsequent experiments by setting the probability parameter $\sigma$ to 0.

*Text Embedding*: Exp.2 $\sim$ 5 validate the effectiveness of the text embedding module. In Exp.2, captions are encoded directly by the CLIP text encoder. Exp.3 employs the CLIP vision encoder(vit) as the feature extractor, but yielded unsatisfactory results, likely due to images with only rendered text falling into the Out-of-Distribution data category for the pre-trained CLIP vision model, hindering its text stroke encoding capabilities. Exp.4 experimented with a trainable module composed of stacked convolutions and an average pooling layer(conv), which struggled without tailored supervision. In Exp.5, utilizing the pre-trained OCR model(PP-OCRv3) resulted in a significant 25.7% increase in Sen. Acc metric compared to Exp.2.

*Position*: Comparing Exp.5 and Exp.6, although the rendered text in $l_g$ contains positional information implicitly, the inclusion of a more accurate position $l_p$ further improves performance and enables the model to generate text in irregular regions.

*Text Perceptual Loss*: Exp.7 $\sim$ 9 validate the effectiveness of text perceptual loss. Upon conducting three experiments, we found that $\lambda = 0.01$ yielded the best results, with a 4.0% improvement compared to Exp.5 in Sen. Acc metric. It is worth noting that perceptual loss necessitates transitioning from latent space to image space, which slows down the training process. Therefore, for Exp.7 $\sim$ 9, we started training from epoch 13 of Exp.5 and trained only for the remaining 2 epochs. Despite this, perceptual loss demonstrated significant improvements.

## 6 CONCLUSION AND LIMITATIONS

In this paper, we delve into the extensively researched problem of text generation in the field of text-to-image synthesis. To address this challenge, we propose a novel approach called AnyText, which is a diffusion-based multi-lingual text generation and editing framework. Our approach incorporates an auxiliary latent module that combines text glyph, position, and masked image into a latent space. Furthermore, in the text embedding module, we leverage an OCR model to extract glyph information and merge it with the semantic details of the image caption, thereby enhancing the consistency between the text and the background. To improve writing accuracy, we employ text-control diffusion loss and text perceptual loss during training. In terms of training data, we present the AnyWord-3M dataset, comprising 3 million text-image pairs in multiple languages with OCR annotations. To demonstrate the superior performance of AnyText, we conduct comprehensive experiments on the proposed AnyText-benchmark, showcasing its superiority over existing methods. Moving forward, our future work will focus on exploring the generation of extremely small fonts and investigating text generation with controllable attributes.

## 7 REPRODUCIBILITY STATEMENT

To ensure reproducibility, we have made the following efforts: (1) We will release our code and dataset. (2) We provide implementation details in Sec. 5.1, including the training process and selection of hyper-parameters. (3) We provide details on dataset preparation and evaluation metric in Sec 4 and Appendix A.3, and the code and data will be made available along with it.

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

# A APPENDIX

## A.1 MORE EXAMPLES ON THE FLEXIBILITY OF ANYTEXT

By adopting the architecture of ControlNet, AnyText exhibits great flexibility. On one hand, the base model's conventional text-to-image generation capability is preserved as shown in Fig. 7. On the other hand, the open-source community, such as HuggingFace and CivitAI, has developed a wide range of base models with diverse styles. By modularly binding AnyText, as depicted in Fig. 8, these models(Oil Painting[4], Guoha Diffusion[5], Product Design[6], and Moon Film[7]) can acquire the ability to generate text. All of these aspects highlight the highly versatile use cases of AnyText.

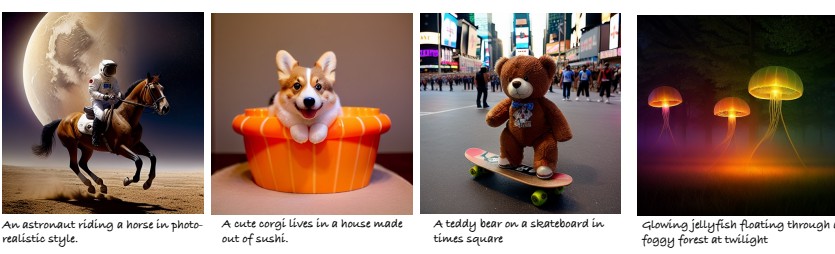

Figure 7: Examples of AnyText generating images without text.

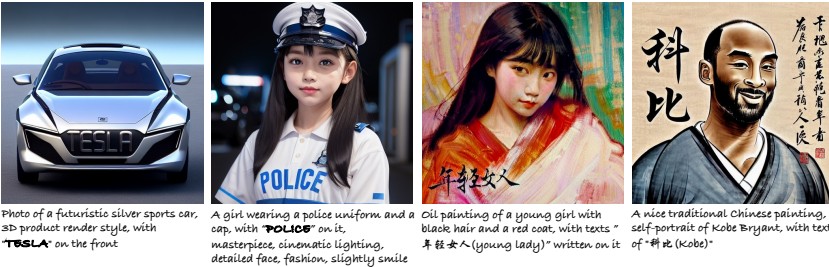

Figure 8: Examples of community models integrated with AnyText that can generate text.

---

[4] https://civitai.com/models/20184

[5] https://civitai.com/models/33/guoha-diffusion

[6] https://civitai.com/models/23893

[7] https://civitai.com/models/43977

Table 4: The Comparison of the parameter sizes of modules between ControlNet and AnyText.

| Modules | ControlNet | AnyText |
|---|---|---|
| UNet | 859M | 859M |
| VAE | 83.7M | 83.7M |
| CLIP Text Encoder | 123M | 123M |
| ControlNet | 361M | - |
| TextControlNet | - | 360M |
| Glyph Block | - | 0.35M |
| Position Block | - | 0.04M |
| Fuse Layer | - | 0.93M |
| OCR Model | - | 2.6M |
| Linear Layer | - | 2.0M |
| Total | 1426.7M | 1431.6M |

## A.2 PARAMETER SIZE AND COMPUTATIONAL OVERHEAD OF ANYTEXT

Our framework is implemented based on ControlNet. Despite the addition of some modules, it did not significantly increase the parameter size, as refered to Table 4. We compared the computational overhead of both models using a batch size of 4 on a single Tesla V100, the inference time for ControlNet is 3476 ms/image, and for AnyText is 3512 ms/image.

## A.3 MORE DETAILS ABOUT DATASET PREPARATION

**Filtering Rules**: We have established strict filtering rules to ensure the quality of training data. Taking the Chinese dataset Wukong Gu et al. (2022) as an example, each original image undergoes the following filtering rules:

- Width or height of the image should be no less than 256.
- Aspect ratio of the image should be between 0.67 and 1.5.
- Area of the text region should not be less than 10% of the entire image.
- Number of text lines in the image should not exceed 8.

Next, for the remaining images, each text line is filtered based on the following rules:

- Height of the text should not be less than 30 pixels.
- Score of OCR recognition of the text should be no lower than 0.7.
- Content of the text should not be empty or consist solely of whitespace.

We implemented similar filtering rules on the LAION-400M Schuhmann et al. (2021) dataset, although the rules were more stringent due to the abundance of English data compared to Chinese. By implementing these rigorous filtering rules, we aim to ensure that the training data consists of high-quality images and accurately recognized text lines.

**Image Captions**: We regenerated caption for each image using BLIP-2 Li et al. (2023) and removed specific placeholders $S_*$ (such as '*'). Then, we randomly selected one of the following statements and concatenated it to the caption:

- ", content and position of the texts are "
- ", textual material depicted in the image are "
- ", texts that say "
- ", captions shown in the snapshot are "
- ", with the words of "
- ", that reads "
- ", the written materials on the picture: "
- ", these texts are written on it: "

Table 5: Statistics of dataset size and line count in subsets of AnyWord-3M.

| Subsets | image count | image w/o text | line count | mean lines/img | #img <=5 lines |
|---|---|---|---|---|---|
| Wukong | 1.54M | 0 | 3.23M | 2.10 | 1.51M, 98.1% |
| LAION | 1.39M | 0 | 5.75M | 4.13 | 1.03M, 74.0% |
| OCR datasets | 0.1M | 21.7K | 203.5K | 2.03 | 93.7K, 93.6% |
| Total | 3.03M | 21.7K | 9.18M | 3.03 | 2.64M, 86.9% |

Table 6: Statistics of characters or words for different languages in AnyWord-3M.

| Lanuages | line count | chars/words count | unique chars/words | mean chars/line | #line <=20 chars |
|---|---|---|---|---|---|
| Chinese | 2.90M | 15.09M | 5.9K | 5.20 | 2.89M, 99.5% |
| English | 6.27M | 6.35M | 695.2K | 5.42 | 6.25M, 99.7% |
| Others | 11.7K | 59.5K | 2.1K | 5.06 | 11.7K, 100% |
| Total | 9.18M | 21.50M | 703.3K | 5.35 | 9.15M, 99.6% |

- ", captions are "
- ", content of the text in the graphic is "

Next, based on the number of text lines, the placeholders will be concatenated at the end to form the final textual description, such as: "*a button with an orange and white design on it, these texts are written on it: *, *, *, **". After processing through the text embedding module, the embeddings corresponding to the placeholders will be replaced with embeddings of the text's glyph information.

## A.4 STASTIC AND EXAMPLES OF ANYWORD-3M

In Table 5 and Table 6, we provide detailed statistics on the composition of AnyWord-3M dataset. Additionally, in Fig. 9, we present some example images from the dataset.

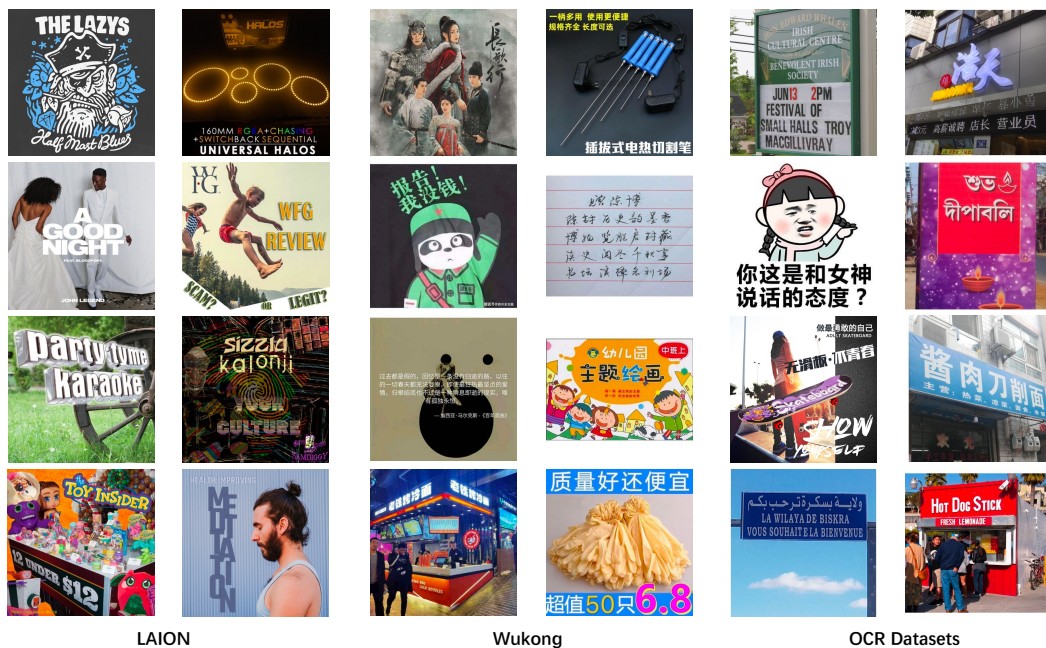

Figure 9: Examples of images from the AnyWord-3M dataset.

## A.5 EXAMPLES FROM ANYTEXT-BENCHMARK

We selected some images from the generated image of AnyText-benchmark evaluation set. The English and Chinese examples can be seen in Fig. 10 and Fig. 11, respectively. All the example images were generated using the same fixed random seed, as well as the same positive prompt ("*best quality, extremely detailed*") and negative prompt ("*longbody, lowres, bad anatomy, bad hands, missing fingers, extra digit, fewer digits, cropped, worst quality, low quality, watermark*").

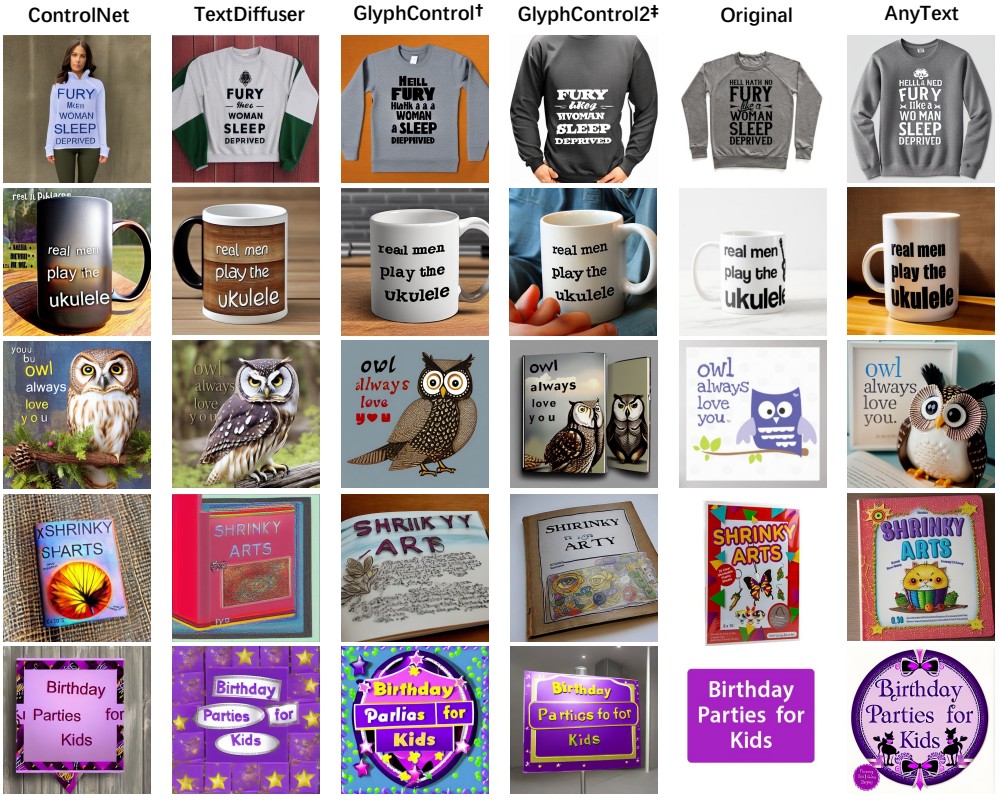

Figure 10: Examples of images in English from the AnyText-benchmark. †is trained on LAION-Glyph-10M, and ‡is fine-tuned on TextCaps-5k.

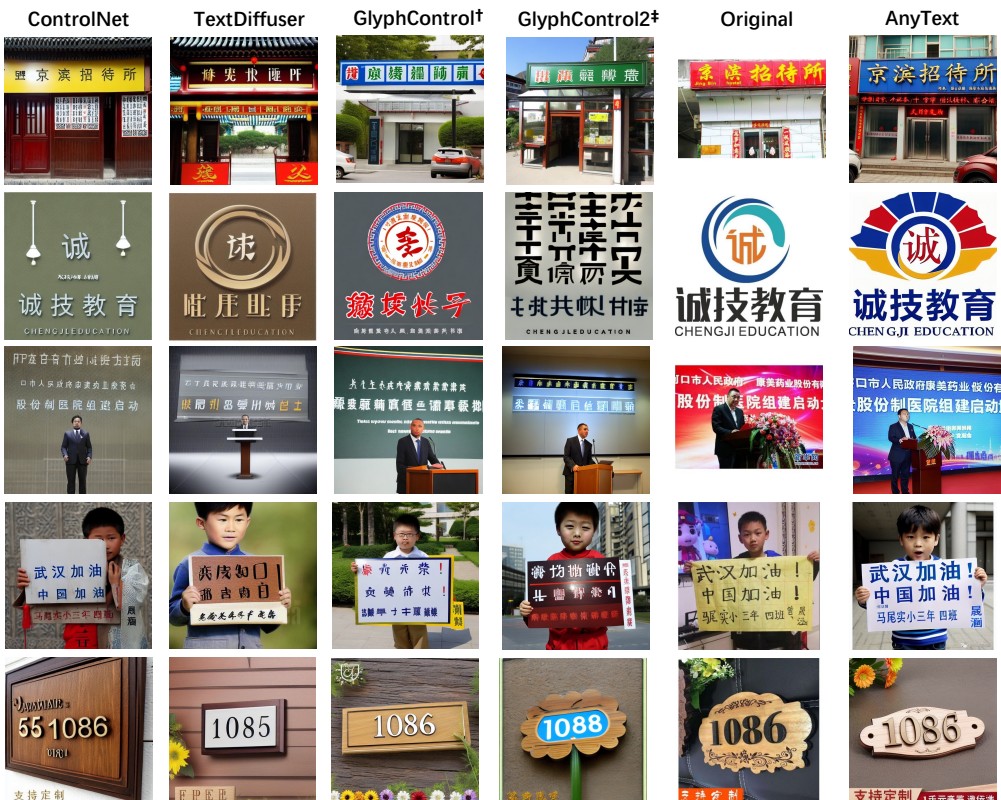

Figure 11: Examples of images in Chinese from the AnyText-benchmark. †is trained on LAION-Glyph-10M, and ‡is fine-tuned on TextCaps-5k.

Table 7: Changes in the data scale of AnyWord-3M.

| Version | Wukong | LAION | Total | wm_score<0.5 |
|---------|--------|-------|-------|--------------|
| v1.0 | 1.54M | 1.39M | 3.03M | – |
| v1.1 | 1.71M | 1.72M | 3.53M | 2.95M |

## A.6   MORE EXAMPLES OF ANYTEXT

We present additional examples in the text generation (see Fig. 12) and text editing (see Fig. 13).

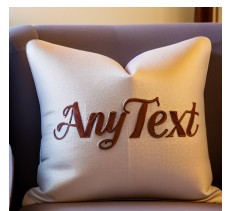
a beautifully crafted decorative pillow with the words "**AnyText**"

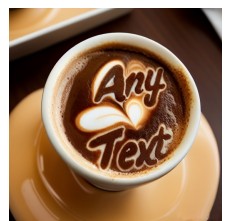
photo of a caramel macchiato coffee, top-down perspective, with "**Any**" "**Text**" written using chocolate

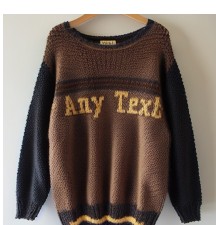
a sweater with knitted text on it, "**AnyText**"

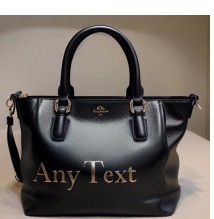
a luxurious women's leather handbag, engraved with the golden words "**AnyText**".

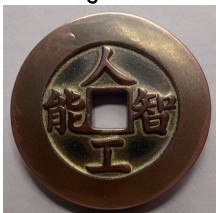
ancient Chinese copper coin, texts on it are "人" "工" "智" "能"(artificial intelligence)

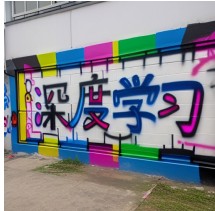
a colorful graffiti on the building, with the words "深度学习"(deep learning)

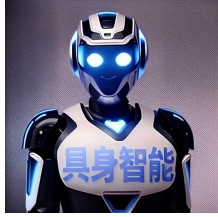
a futuristic-looking robot, with the words "具身智能"(embodied AI) on his body

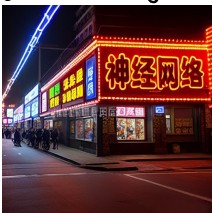
on a bustling street during the night, neon lights spell out "神经网络"(neural network)

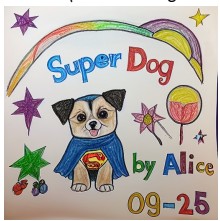
a nice drawing by crayons, a dog wearing a superhero cape soars sky, with the words "**SuperDog**" "**by Alice**" and "**09-25**"

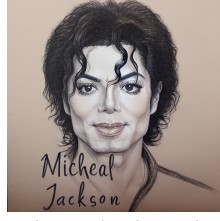
a nice drawing in pencil of Michael Jackson, with the words "**Micheal**" and "**Jackson**"

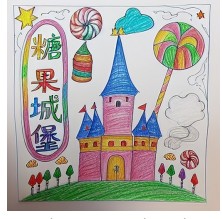
a drawing by a child with crayons, a castle made of lollipops, marshmallows, and candies, "糖果城堡(candy castle)"

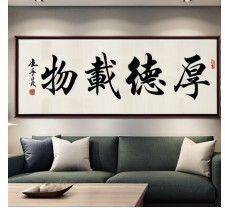
nice Chinese calligraphy work hanging in the middle of the living room, with words "物载德厚(self-discipline and social commitment)"

Figure 12: More examples of AnyText in text generation.

## A.7   IMPLEMENTATION DETAILS OF ANYTEXT-V1.1

We randomly sampled some text lines in AnyWord-3M and observed that the OCR annotation error rate for the English portion was 2.4%, while for the Chinese portion was significantly higher at 11.8%. So, for the Chinese part, we used the latest PP-OCRv4 PaddlePaddle (2023) to regenerate the OCR annotations; and for the English part, we found that the annotations from MARIO-LAION by TextDiffuser Chen et al. (2023b) performed slightly better on some severely deformed English sentences compared to PP-OCRv4, so we replaced them with their annotations. Additionally, we slightly increased some data and making the ratio of the English and Chinese was approximately 1:1. This updated dataset was labeled as v1.1, as shown in Table 7. In addition, we added a "wm_score" label for each image to indicate the probability of containing a watermark, which is used to filter images with watermarks in the final stage of model training.

Furthermore, we proposed a method called "inv_mask". For each text line in the image, if its recognition score is too low, or the text is too small, or if it is not among the 5 randomly chosen text lines,

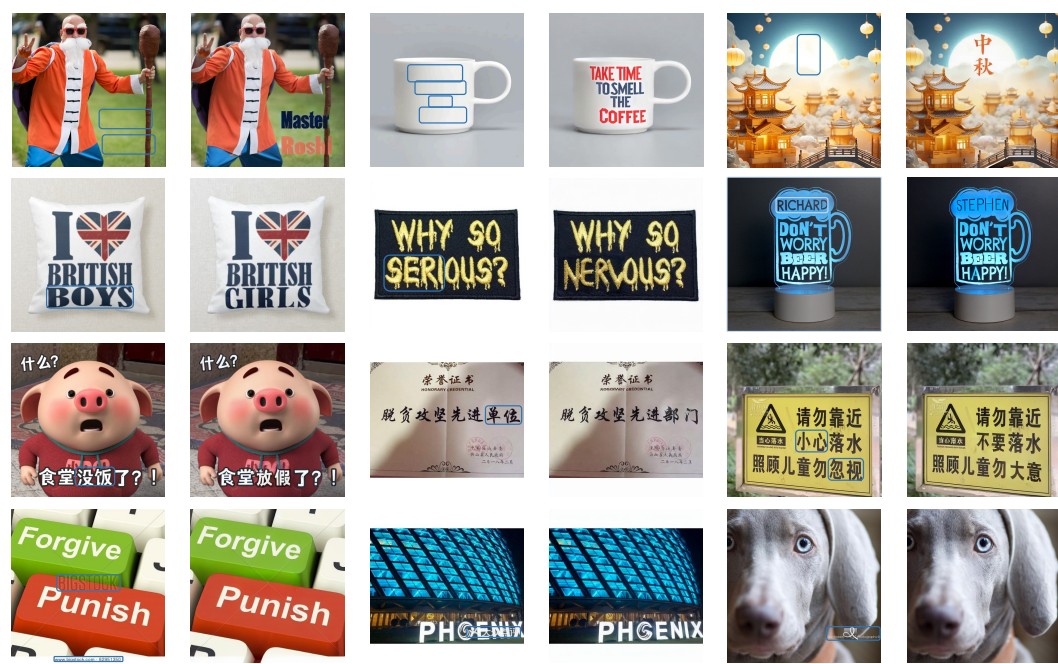

Figure 13: More examples of AnyText in text editing.

Table 8: Improvement of watermark and pseudo-text of AnyText, tested on private model and dataset.

| Version | English | | Chinese | |
|---|---|---|---|---|
| | watermark | pseudo-text | watermark | pseudo-text |
| v1.0 | 6.9% | 119.0% | 24.7% | 166.8% |
| v1.1 | 0.4% | 110.9% | 2.9% | 124.4% |

it will be marked as invalid. Subsequently, these invalid text lines are combined to form a mask, and during training, the loss in the corresponding area will be set to 0. This straightforward approach can bring significant improvement in OCR metrics, as illustrated in Table 2, and also notably reduced the ratio of pseudo-text areas in the background, as depicted in Table 8.

The v1.0 model underwent further fine-tuning on the AnyWord-3M v1.1 dataset for 5 epochs, with the last 2 epochs using data of wm_score $< 0.5$, filtering out approximately 25% Chinese data and 8% English data. The parameter $\lambda$ for perceptual loss was set to 0.003, as it yielded favorable results for both Chinese and English on the v1.1 dataset. Subsequent to the training process, the original SD1.5 base model was replaced with a community model *Realistic_Vision*[8], which makes the generated images more aesthetically appealing. The final model was labeled as version v1.1.

---

[8]https://huggingface.co/SG161222/Realistic_Vision_V5.0_noVAE

