# OpenReview forum: "AnyText: Multilingual Visual Text Generation and Editing"
_ICLR.cc/2024/Conference — ICLR 2024 spotlight_

### Official Review · Reviewer_LoJV · 2023-10-29

**Soundness:** 3 good
**Presentation:** 3 good
**Contribution:** 2 fair
**Rating:** 6
**Confidence:** 3

**Summary:**

This paper proposed AnyText, a diffusion-based multilingual visual text generation and editing model. It combines auxiliary latent model which is a control net for text condition, and a text embedding module which injects text visual information in the prompt latent space. Text-control diffusion loss and text perceptual loss are using in training. A large-scale multilingual text images dataset, AnyWord-3M, is introduced.

**Strengths:**

- extended control net for text input condition
- new visual text token embedded in the prompt
- introduced OCR related perceptual loss
- new dataset
- new state-of-the-art under proposed new evaluation benchmark

**Weaknesses:**

- using models trained own dataset to compare with previous baselines is not so fair
- the requirement of user given text mask is not always easy in practice.

**Questions:**

- It is not clear whether the improved results come from better training data or the proposed model. It would be best to compare the baseline models trained on the same dataset, or train the proposed model on the previous LAION glyph subset.
- in the experiments, the ground truth text mask is used as conditional input. It would be interesting to see what if random text position and mask is used, can it still generate reasonable image?

---

> ### Author Response · Authors · 2023-11-20
> **Replay to Reviewer LoJV**
>
> Dear Reviewer,
>
> We deeply appreciate your thorough and meticulous review, and we would like to express our sincere gratitude for recognizing the value of our work. Your thoughtful questions have been greatly appreciated, and we are more than willing to provide detailed explanations in response:
>
> >Q1: It is not clear whether the improved results come from better training data or the proposed model. It would be best to compare the baseline models trained on the same dataset, or train the proposed model on the previous LAION glyph subset.
>
> The training dataset used for TextDiffuser is MARIO-10M, out of which 9.2M images were filtered from LAION-400M. Similarly, the English data in AnyWord-3M was also filtered from LAION-400M. During the data processing, we indeed found that there were nearly 10 million images containing text in LAION-400M. However, in order to maintain a balance in terms of scale with the Chinese data in AnyWord-3M, we adjusted our filtering rules and retained only 1.3M images. More details about this can be found in Appendix A.2. To some extent, we are using a subset of the training data as for TextDiffuser.
>
> The GlyphControl dataset, LAION-Glyph-10M, was filtered from LAION-2B-en, and it is likely to have a significant overlap with the text-containing images in LAION-400M. Additionally, in our ablation experiments (refer to Table 2 in Supplementary Material for the latest results), the setup of Exp. 2 is very similar to GlyphControl. It involves disabling editing, text embedding, and perceptual loss while retaining only the glyph image that renders tests based on their positions, along with the corresponding text position image. Under the exact same training dataset, our method (Exp. 8) demonstrates a significant advantage.
>
> In addition, we randomly sampled some text lines to assess the accuracy of our OCR annotations. The error rate for the English portion was 2.4%, while for the Chinese portion was significantly higher at 11.8%. This difference may be attributed to the inherent complexity of Chinese character recognition, as well as the performance of PPOCR_v3.
>
> In summary, during model training, we only utilized 1.3 million out of 10 million English data and incorporated over 1.6 million non-English data with noise. While our English metrics have a clear advantage over competing methods, this advantage is certainly not derived from the benefits of the data.
>
> >Q2: in the experiments, the ground truth text mask is used as conditional input. It would be interesting to see what if random text position and mask is used, can it still generate reasonable image?
>
> The ground truth text mask is only available during evaluation. In other cases, there is no ground truth mask. In such cases, the user needs to manually specify the position of the text, as shown in Fig. 1 in Supplementary Material, using tools like the Gradio brush. It is true that providing a mask may not be very user-friendly, and if the position of the text is too arbitrary or unreasonable, it can also decrease the quality of the generated image. We also provide a template-based approach to automatically generate masks. However, it is important to acknowledge that the ability for users to specify the position of the text is indispensable. In certain situations, relying solely on methods that automatically predict text positions may not fully meet all users' requirements.

---

> > ### Comment · Reviewer_LoJV · 2023-11-20
> >
> > Using a small training set does not necessary mean a disadvantage, given LAION annotation is noisy. Even this is true, it is still necessary to see a fair comparison for how much improvement is obtained solely by algorithm.
> >
> > It seems other methods using large training set will inevitably get trained on the evaluation set used by this paper. It is amazing even in this way they underperform the proposed approach. This is not normal. Can training on Chinese data also contribute to improvement on English?
> >
> > I have more doubts after considering authors' reply.

---

> > > ### Comment · Reviewer_LoJV · 2023-11-20
> > >
> > > It's understood retraining on the same set is prohibitive. How about evaluating authors' model on the evaluation set used by other work and using the same metric? This should be easy to do.

---

> ### Author Response · Authors · 2023-11-21
> **Replay to Reviewer LoJV**
>
> Dear reviewer,
>
> We attempted to compare our model with the method and data used in another work, specifically TextDiffuser. However, we encountered some challenges that made it not as "easy to go" as expected. Following the instructions provided in the [Official Code](https://github.com/microsoft/unilm/blob/master/textdiffuser), we found that it only provides rendered images with all text lines. Since AnyText requires the position and corresponding text for each line to make the Auxiliary Latent and Text Embedding modules work, perform inference became impossible. To overcome this issue, we attempted an alternative approach by using our OCR model to generate annotations, followed by generating images and comparing them using their metrics. After spending lots of time configuring the OCR model they used ([MaskTextSpotterV3](https://github.com/MhLiao/MaskTextSpotterV3)), and made guesses for missing details (presumably modified in the MaskTextSpotterV3 code that was not provided), we roughly replicated their evaluation metrics. Unfortunately, we encountered another issue that our OCR results are not consistent with the ground truth they used in their precision and recall calculations, which prevent us from obtaining reliable and fair evaluation results. Regarding GlyphControl, we were even unable to reproduce their own evaluation results. From the [Official Code](https://github.com/AIGText/GlyphControl-release/tree/main), they only shared 200+ prompt templates used for evaluation, without providing the texts they randomly selected and the positions of them, and the evaluation code was not provided, either. Consequently, we are unable to use evaluation set and metric from others to assess our model.
>
> In the other question, it is worth considering that other methods might have seen some of our evaluation sets. However, even that if a generated model has seen an image during training, fully restoring the pixels accurately is still challenging. Furthermore, the inputs provided to the model during inference are not the actual images themselves but rather captions and OCR annotations, which could differ from those used by other models. So, we believe that compared to traditional methods such as detection and recognition, the impact of this issue is relatively minor.
>
> Regarding whether the presence of Chinese data improves the performance of English text, our answer is: very likely. Chinese characters encompass thousands of distinct shapes, and if a model can accurately generate these characters, it becomes comparatively much easier to generate the additional 26 concise letterforms of English. This stems from our method's comprehensive approach to designing methods and processing datasets to support multilingual capabilities, which may not be present in other methods. However, even when considering the same training data, if we remove $l_p$ and $l_m$ in the Auxiliary Latent module, disable the Text Embedding module, and exclude the Text Perceptual loss, our method bears resemblance to GlyphControl. In our ablation study (see Table 2 in Supplementary Materials for latest), we have demonstrated performance improvements achieved by each of these self-designed algorithms.
>
> We hope this reply has addressed your concerns, and we are looking forward to further discussion with you. Thank you!

---

### Official Review · Reviewer_5GTp · 2023-10-31

**Soundness:** 3 good
**Presentation:** 3 good
**Contribution:** 3 good
**Rating:** 6
**Confidence:** 3

**Summary:**

This paper introduces modules to enhance the text-drawing capabilities of text-to-image diffusion models. The auxiliary latent module embeds glyph and position information obtained from an off-the-shelf OCR module and fuses these latents as diffusion-step invariant conditions through ControlNet. Additionally, the text embedding module encodes a tokenized prompt, replacing the tokens of rendered glyphs with special tokens. Since these special tokens, the components of the auxiliary latent module, and ControlNet are the only trainable parts in the entire model, this method can be readily applied to existing diffusion models without retraining the diffusion UNet. The modules are trained on the AnyWord-3M dataset, also proposed in this paper. The performance of the proposed method surpasses that of previous text-generation-focused text-to-image diffusion models and also offers multilingual text generation capabilities.

**Strengths:**

Generating text glyphs properly in images produced by text-to-image diffusion models has been a longstanding issue. Research has shown that this capability can be improved by increasing data and model size, but this is somewhat obvious or expected. Following ControlNet, which proposes controllability for Diffusion UNet, the text glyph generation problem can be solved; however, as shown in this paper, it would result in a monotonous style. One of the paper's strengths is that the generated glyphs harmonize with the content of the generated images and are not monotonous. Additionally, the paper's ability to handle multilingual text glyph generation with relatively less data is another notable strength.

**Weaknesses:**

As revealed in the ablation study, the most significant performance improvement of this method occurs upon the introduction of text embedding. This is attributed to the performance of PP-OCRv3. If one were not to use the OCR module's embedding and instead employ a general image encoder like the CLIP visual encoder, it is questionable whether the same level of performance improvement would have been achieved. Additionally, many modules are added to the vanilla text-to-image diffusion model, but the paper fails to mention the computational overhead that arises as a result. Although the paper highlights multilingual capabilities and provides qualitative results for Korean and Japanese in the figures, it is disappointing that these two languages are excluded from the quantitative results, falling under the "others" category. Furthermore, it is regrettable that the results of the ablation study are listed only in terms of Chinese sentence accuracy and NED, without any FID measurements. The lack of qualitative results corresponding to each ablation experiment is also a drawback.

**Questions:**

I'd like to pose questions that can address the weaknesses discussed.

1. Could you elaborate on how the PP-OCRv3's performance specifically influences the results?
2. Have you considered measuring the computational overhead when additional modules are integrated into the vanilla text-to-image diffusion model?
3. Why were Korean and Japanese languages included in the qualitative results but not in the quantitative ones?
4. Is there a reason why FID measurements were not included in the ablation study?
5. Why were qualitative results not provided for each individual ablation experiment?

---

> ### Author Response · Authors · 2023-11-20
> **Replay to Reviewer 5GTp**
>
> Dear Reviewer,
>
> We express our heartfelt gratitude for your thorough review of our manuscript and the recognition it has received. We have given careful consideration to the issues you raised and offer our responses below, and we would like to engage in further discussion.
>
> >Q1: Could you elaborate on how the PP-OCRv3's performance specifically influences the results?
>
> Coincidentally, before utilizing the OCR model, our initial attempt was to employ the CLIP visual encoder. However, we found that the results were unsatisfactory. We suspect that this is because images with only rendered text fall under the Out-of-Distribution data category for the pre-trained CLIP vision model, hindering its text stroke understanding and encoding capabilities. Subsequently, we experimented with a trainable module composed of stacked convolutions and an average pooling layer. However, without strong supervision specifically tailored to this task, this module also struggled to perform well. In this study, we have reorganized our experiments, and the relevant results can be found in Table 2 of Supplementary Material. From Exp. 3 and Exp. 4, we noticed that using the vit or conv module for token replacement in the text embedding resulted in even lower performance compared to the original CLIP text encoder in Exp .2. Comparing with Exp. 5, we can conclude that utilizing a pre-trained OCR model effectively encodes stroke-level information, significantly enhancing text generation accuracy. You can also find qualitative comparisons in Fig. 4 and Fig. 5 in Supplementary Material.
>
> >Q2: Have you considered measuring the computational overhead when additional modules are integrated into the vanilla text-to-image diffusion model?
>
> As mentioned in the paper, our framework is based on ControlNet. Despite the addition of some modules, it did not significantly increase the computational overhead. Please refer to the parameter details below:
> Params | ControlNet | AnyText
> :---:|:---:|:---:
> UNet|859M|859M
> VAE|83.7M|83.7M
> CLIP Text Encoder|123M|123M
> ControlNet|361M|-
> TextControlNet|-|360M
> Glyph Block|-|0.35M
> Position Block|-|0.04M
> Fuse Layer|-|0.93M
> OCR Model|-|2.6M
> Linear Layer|-|2.0M
> Total|1426.7M|1431.6M
>
> We compared the computational overhead of both models using a batch size of 4 on a single Tesla V100. The results are as follows:
> Model | Inference Time
> :---:|:---:
> ControlNet|3476ms/image
> AnyText|3512ms/image
>
> In summary, compared to ControlNet, AnyText has an increase of 0.34% in parameter size and 1.04% in inference time.
>
> >Q3: Why were Korean and Japanese languages included in the qualitative results but not in the quantitative ones?
>
> Generating quantitative results for these languages requires the careful selection of reliable OCR models and the meticulous preparation of benchmark datasets. However, to be honest, none of us are familiar with languages other than Chinese and English. Furthermore, even if we were to calculate quantitative results, there are currently no comparable methods available to the public. Therefore, we have only provided qualitative results for now. Nevertheless, we plan to gradually improve this aspect in the future.
>
> >Q4: Is there a reason why FID measurements were not included in the ablation study?
>
> Due to limitations in computational resources, we were unable to conduct ablation experiments on the full dataset and instead opted for a smaller-scale dataset. At this scale, the generated images from the ablation experiments did not exhibit satisfactory realism, making the FID metric less meaningful. In contrast, our primary focus was on the accuracy of the generated text, measured by Sen.ACC and NED. We plan to increase the scale of the ablation experiments in the future.
>
> >Q5: Why were qualitative results not provided for each individual ablation experiment?
>
> We have added qualitative results in Fig. 4 and Fig. 5 in Supplemental Material.

---

> > ### Comment · Reviewer_5GTp · 2023-11-22
> >
> > Dear authors,
> >
> > Thank you for your detailed reply to my questions.
> > I have read your revised supplementary material, and it cleared most of my doubts.
> >
> > Overall, it seems a major improvement and the contribution came from employing OCR backbone as double-checked in my question #1 and table 2 in the supplementary material.
> > In my opinion, my initial rating of "6" seems adequate as papers like [1] have already shown the efficacy of employing OCR backbone for generative models (thus, the novelty is somewhat limited).
> >
> > [1] Rodriguez, Juan A., et al. "OCR-VQGAN: Taming text-within-image generation." Proceedings of the IEEE/CVF Winter Conference on Applications of Computer Vision. 2023.

---

### Official Review · Reviewer_xCSM · 2023-10-31

**Soundness:** 3 good
**Presentation:** 3 good
**Contribution:** 4 excellent
**Rating:** 8
**Confidence:** 4

**Summary:**

The manuscript unfolds AnyText, a profound diffusion-based multilingual visual text generation and editing model. It meticulously tackles the intricacies involved in precise text portrayal within generated images, deploying auxiliary latent modules and text embedding modules as strategic tools. To augment the training phase, the introduction of text-control diffusion loss and text perceptual loss is articulated, which serves to bolster text generation quality. A formidable performer, AnyText triumphs over existing paradigms, championing improved accuracy and quality in text generation. Furthermore, the introduction of a novel dataset, AnyWord-3M, enriches the existing reservoir of multilingual image-text pairs, reflecting a thoughtful contribution to the scholarly community.

**Strengths:**

(1) A notable innovation lies in the paper's strategic approach to circumvent challenges, ensuring precise text portrayal within generated images.

(2) The infusion of auxiliary latent modules coupled with text embedding modules acts as a catalyst, promoting enhanced accuracy and coherence in the text generation process.

(3) Strategic incorporation of text-control diffusion loss and text perceptual loss during training heralds improvement in the overall text generation quality.

(4) A commendable addition is the introduction of AnyWord-3M, a robust dataset enriched with 3 million image-text pairs, elaborately annotated with OCR in multiple languages, signifying a valuable asset to the research fraternity.

**Weaknesses:**

(1) The architecture seems somewhat reliant on pre-established technologies such as Latent/Stable Diffusion and ControlNet, which slightly shadows its novelty.

(2) Encumbered by a complex array of components and a multi-stage training regimen, the model’s re-implementation emerges as a challenging task, compounded further by numerous critical hyperparameters requiring manual assignment.

(3) Certain aspects, such as token replacement, require a more elaborate discourse for clearer comprehension, primarily concerning the identification of corresponding tokens and their subsequent utility in text-image generation.

(4) There exists a potential ambiguity concerning the intermediate generative results (x'_0), where the possible presence of noise or blur could compromise the precision of perceptual loss computation.

(5) A clearer depiction of computational resource demands (GPU Hours), beyond the ambiguity of epochs, would enhance the paper’s practicability and replicability.

(6) A more explicit elucidation on the operational synergy between Z_a from the Auxiliary Latent Module and Z_0 from VAE, as depicted in Fig. 2, alongside their application within Text-ControlNet, would augment the manuscript's clarity.

**Questions:**

N/A

---

> ### Author Response · Authors · 2023-11-20
> **Replay to Reviewer xCSM - Part 1/2**
>
> Dear reviewer,
>
> Thank you very much for your affirmation and praise of our work， which has greatly encouraged us, and we will continue to make efforts to open-source and further improve AnyText, advancing the development of text generation technology. Regarding the issues you mentioned, our responses are as follows:
>
> >Q1: The architecture seems somewhat reliant on pre-established technologies such as Latent/Stable Diffusion and ControlNet, which slightly shadows its novelty.
>
> Stable Diffusion and ControlNet have been gaining significant popularity in the open-source community recently, and various interesting models have emerged. Our intention was to provide a plug-and-play module that allows more people to work on text generation and editing based on their own models. However, technically speaking, pre-established technologies are not strict dependencies for AnyText. In the future, we may explore other methods such as GANs.
>
> >Q2: Encumbered by a complex array of components and a multi-stage training regimen, the model’s re-implementation emerges as a challenging task, compounded further by numerous critical hyperparameters requiring manual assignment.
>
> Text generation and editing can be seen as a multi-task model. It is logical to first focus on "generating" text well before proceeding to "editing" it, which is why we implemented a staged training approach. The current solution does involve several modules and parameters, which is why we built a smaller-scale dataset (mentioned in Sec. 5.3) and conducted extensive validation work. In the future, we will open-source our code and dataset to facilitate the reproducibility of AnyText and consider refining the approach to make it more streamlined.
>
> >Q3: Certain aspects, such as token replacement, require a more elaborate discourse for clearer comprehension, primarily concerning the identification of corresponding tokens and their subsequent utility in text-image generation.
>
> The process of token replacement is as follows: First, we obtain the CLIP token index of the placeholder string. For example, the index of '$\*$' is 265. Then, for the input $y'$ (which is processed from the original caption $y$ with all text lines replaced by '$*$'), after tokenization, we obtain a sequence of token indices. At this point, we keep track of the positions where the index is 265. After obtaining all token embeddings through a lookup table, we replace the embeddings at the marked positions, with the embeddings encoded by the OCR model. The replaced tokens then go through the self-attention mechanism of the CLIP text encoder, which combines stroke information with semantic information.
>
> >Q4: There exists a potential ambiguity concerning the intermediate generative results (x'_0), where the possible presence of noise or blur could compromise the precision of perceptual loss computation.
>
> Yes, the $x'\_{0}$ may appear blurry, and the blurriness is correlated with the time step t. This is why we introduced the weight adjustment function $\varphi(t) = \bar{\alpha_t}$. We provide a more detailed explanation in Fig. 6 and Fig. 7 in Supplementary Material. According to the original DDPM paper, $\alpha_t=1-\beta_t$, where $\beta_t$ represents the forward process variances that increase linearly from 0.0001 to 0.02. $\bar{\alpha_t}$ is the cumprod of $\alpha_t$, with the range from 0 to 1, this aligns perfectly with our requirements for the weight adjustment. When t is large, the predicted image $x'_{0}$ may be more blurry, but the corresponding weight is also small, and the text perceptual loss has little effect.
>
> >Q5: A clearer depiction of computational resource demands (GPU Hours), beyond the ambiguity of epochs, would enhance the paper’s practicability and replicability.
>
> As mentioned in Sec. 5.1, our model was trained for 10 epochs, with the training speed being consistent for the first 8 epochs at approximately 29h/epoch. For the last 2 epochs, it increased to ~40h/epoch due to the enabling of the text perceptual loss. Overall, the total training time for the model amounted to 312 hours on 8XA100 GPUs.

---

> > ### Author Response · Authors · 2023-11-20
> > **Replay to Reviewer xCSM - Part 2/2**
> >
> > >Q6: A more explicit elucidation on the operational synergy between Z_a from the Auxiliary Latent Module and Z_0 from VAE, as depicted in Fig. 2, alongside their application within Text-ControlNet, would augment the manuscript's clarity.
> >
> > In the standard stable diffusion process, training the model on $z_t$ alone cannot generate text well, since it does not explicitly inform the model of how (*glyphs*) and where (*position*) to generate them. Therefore, we designed the Auxiliary Latent Module to combine the information into $z_a$. As you mentioned, we hoped to use $z_a$ as a "catalyst" for text generation by ensuring that its shape aligns with $z_t$ and adding them together. However, we did not directly input this combined latent into the UNet because the parameters of UNet are trained on billions of images, giving it a large knowledge capacity. We did not want to influence its parameters using our relatively smaller dataset of millions of text-containing images. Hence, we freeze these parameters and fed $z_a+z_t$ into an additional trainable TextControlNet, essentially patching the original model to enable text generation.

---

> ### Comment · Reviewer_xCSM · 2023-11-23
> **After Rebuttal**
>
> Thanks for the hard work of authors during rebuttal. This is indeed a high quality paper with good performance as demonstrated in the paper. It would be great that authors could open-source the code, models and dataset to contribute the community in the near future. No more questions and will maintain my initial score.

---

### Official Review · Reviewer_TuoG · 2023-11-01

**Soundness:** 2 fair
**Presentation:** 2 fair
**Contribution:** 2 fair
**Rating:** 6
**Confidence:** 3

**Summary:**

This paper presents an adapter-based module that can be plugged into existing diffusion models to perform multilingual visual text generation and editing. It contains a control net to control the text location and text content and a special text embedding module to improve the multilingual text generation ability. This paper also presents a large-scale multilingual text image dataset, AnyWord-3M, with image-text pairs and OCR annotations in multiple languages.

**Strengths:**

1. The proposed adapter-based module is a plug-and-play module that can guide visual text generation of many existing pre-trained diffusion models and can apply to multiple languages, which was not achieved in previous works.
2. The proposed text adapter and text encoder are proven to be effective in improving the OCR performance.
3. The proposed AnyWord-3M dataset is the first large-scale text image dataset with multilingual OCR annotations and is useful for future study.

**Weaknesses:**

1. The method and dataset collection miss a lot of details. For example, is the linear projection layer trained or fixed for ocr encoder in the text embedding module?
2. A lot of information is not presented in the examples shown in the paper, for example, the layouts for images in Figure 1, and the captions for images in Figure 5.

**Questions:**

I have some questions about the model architecture, dataset construction, and experiment design.

For the model architecture:
1. Which feature is extracted from PP-OCR to serve as text embedding?
2. Is the linear projection layer for the OCR encoder trained?
3. What is the configuration for the fuse layer? A single convolution layer or a stacked convotion?
4. What is the input to text controlnet? Based on Figure 2, it seems to be the concatenation of $z_a$ and $z_t$.
5. Why is the image resolution for glyph image 1024x1024 instead of 512x512? This does not align with the final image resolution.

For the dataset construction:
1. How is the tight position mask $l_p$ annotated? As far as I know, the PP-OCR model does not support arbitrary shape text detection.
2. The glyph image $l_g$ contains rotated text; how is the bounding box annotated from the position mask?
3. The captions are generated using BLIP-2 model instead of the original captions. Could authors provide some statistics like the length of the captions and show some example captions for images in AnyWord-3M? How does this difference affect the model performance?

For the experiment:
1. Could authors provide more information about the input for the visual examples? For example, the layouts for Figures 1, 6, 8, 12, the captions for Figures 5, 10, 11.
2. The Sen. ACC and NED metric is measured using the same OCR model as the encoder in model training, which might be unsuitable. Could authors evaluate the OCR performance using another OCR model?
3. Table 3 shows the improvement brought by visual text embedding in Chinese text generation. I wonder if this also improves the English word generation.
4. In the experiment, the text diffuser model is not trained on the same dataset as GlyphControl and AnyText Model. Is it possible that authors fine-tune them on the same data and compare final performance?

---

> ### Author Response · Authors · 2023-11-20
> **Replay to Reviewer TuoG - Part1/2**
>
> Dear reviewer,
>
> We greatly appreciate your time and effort in reviewing our manuscript. Sorry for the oversight of certain details in our paper that have confused you. We are also grateful for the several constructive suggestions which have contributed to the refinement of our work, and we eagerly anticipate further discussions. We have carefully considered each of your concerns and have addressed them thoroughly below:
>
> >Q1: Which feature is extracted from PP-OCR to serve as text embedding?
>
> We use the feature map before the last fully connected layer of the PP-OCR model. The feature map mentioned here and the $\hat{m}\_{p},\hat{m'}\_{p}$ referred to in the text perceptual loss(Sec. 3.4) are the same, the difference is that the feature is flattened and linear projected later, to match the token dimension.
>
> >Q2: Is the linear projection layer for the OCR encoder trained?
>
> Yes, the linear transformation $\xi$ is randomly initialized and trained during the training process.
>
> >Q3: What is the configuration for the fuse layer? A single convolution layer or a stacked convolution?
>
> A single layer, as mentioned in Sec. 3.2 : “...we utilize **a convolutional fusion layer** $f$ to merge $l_g$, $l_p$, and $l_m$...”. The kernel_size=3, stride=1, in_channels=324(256+64+4, channels for $l_g$, $l_p$, and $l_m$) and out_channels=320.
>
> >Q4: What is the input to text controlnet? Based on Figure 2, it seems to be the concatenation of $\textbf{\textit{z}}_a$ and $\textbf{\textit{z}}_t$.
>
> There is a "⊕" in Fig. 2, which means the operation is addition rather than concatenation. To provide an accurate description, we will modify the text in Sec. 3.1 as: "...to control the generation of text, we ~~combine~~ **add** $z_a$ with $z_t$ and fed them into a ..."
>
> >Q5: Why is the image resolution for glyph image 1024x1024 instead of 512x512? This does not align with the final image resolution.
>
> We have observed that when rendering small text onto glyph images at a resolution of 512x512, the text becomes nearly illegible, whereas at 1024x1024 there is no issue (see Supplemental Material Fig. 2, the first and fourth images in the second column). Additionally, we employed 4 downsampling layers for the Glyph block, while the Position block only employed 3. This incurs additional parameter overhead (see our replay to reviewer 5GTp's Q2), but the results demonstrate that it does not affect the alignment with the final image resolution.
>
> >Q6: How is the tight position mask $l_p$ annotated? As far as I know, the PP-OCR model does not support arbitrary shape text detection.
>
> Our dataset does not rely solely on OCR models but also includes human annotations. Just as mentioned in Sec. 4, "Except for the OCR datasets, **where the annotated information is used directly**, all other images are processed using the PP-OCRv3...". The image used in Fig. 2 is sourced from an OCR dataset called Art.
>
> >Q7: The glyph image $l_g$ contains rotated text; how is the bounding box annotated from the position mask?
>
> As mentioned in Sec. 3.2: "...we simplify the process by rendering characters **based on the enclosing rectangle** of the text position...". Specifically, use cv2.minAreaRect( ) on the polygon of the position mask.
>
> >Q8: The captions are generated using BLIP-2 model instead of the original captions. Could authors provide some statistics like the length of the captions and show some example captions for images in AnyWord-3M? How does this difference affect the model performance?
>
> The average caption length by BLIP-2 is **~8.26 words** per image, excluding the additional description of the text to be rendered. You can find caption examples in Fig. 2 and Fig. 3 in Supplemental Material. We utilize BLIP-2 not due to performance considerations, but because images as in OCR datasets or Noah-Wukong do not have existing English descriptions, and BLIP-2 is one of the most common approaches adopted by many studies. We believe that even perfect captions would have negligible influence on text-generation tasks. Therefore, we did not focus on evaluating it.
>
> >Q9: Could authors provide more information about the input for the visual examples? For example, the layouts for Figures 1, 6, 8, 12, the captions for Figures 5, 10, 11.
>
> To provide more information, we have regenerated several images, as shown in Fig. 1 in Supplementary Material. We just use Gradio's brush as input and after a simple process convert it into Glyph and Positon. The layouts and captions for Figures 5, 10, and 11 are available as they are sourced from our benchmark. See Fig. 2 and Fig.3 in Supplemental Material.

---

> > ### Author Response · Authors · 2023-11-20
> > **Replay to Reviewer TuoG - Part2/2**
> >
> > >Q10: The Sen. ACC and NED metric is measured using the same OCR model as the encoder in model training, which might be unsuitable. Could authors evaluate the OCR performance using another OCR model?
> >
> > Thanks for your suggestions. We have chosen a recently popular OCR model in the ModelScope community, DuGuangOCR[1], to replace PPOCR_v3 as our evaluation model, which supports recognition for both Chinese and English. The updated quantitative comparison can be seen in Table 1 in Supplementary Material. We observed that in all methods, there were minimal changes in the Sen.Acc and NED metrics, except for AnyText, which showed a significant decrease. This indicates that PPOCR_v3 indeed introduced bias to the evaluation. However, the updated results still demonstrate that AnyText has significant advantages over competing methods.
> >
> > >Q11: Table 3 shows the improvement brought by visual text embedding in Chinese text generation. I wonder if this also improves the English word generation.
> >
> > Yes, the text embedding works for English. In fact, as we use the feature before the fully connected layer, it is not particularly sensitive to language and instead focuses more on stroke. The Chinese and English evaluation results for Exp. 2 and Exp. 3 in Table 3 are as follows:
> > Exp. № | Text Embedding |Sen. Acc↑(CH) | NED↑(CH) | Sen. Acc↑(EN) | NED↑(EN)
> > :---:|:---:|:---:|:---:|:---:|:---:
> > 2|w/o|0.2024|0.4649|0.3159|0.6298
> > 3|w/|0.4595|0.7072|0.4524|0.7459
> >
> > There is also a qualitative comparison in Fig. 4 in Supplementary Material(Exp. 2 vs. Exp. 5).
> >
> > >Q12: In the experiment, the text diffuser model is not trained on the same dataset as GlyphControl and AnyText Model. Is it possible that authors fine-tune them on the same data and compare final performance?
> >
> > The training dataset used for TextDiffuser is MARIO-10M, out of which 9.2M images were filtered from LAION-400M. Similarly, the English data in AnyWord-3M was also filtered from LAION-400M. During the data processing, we indeed found that there were nearly 10 million images containing text in LAION-400M. However, in order to maintain a balance in terms of scale with the Chinese data in AnyWord-3M, we adjusted our filtering rules and retained only 1.3M images. More details about this can be found in Appendix A.2. To some extent, we are using a subset of the training data as for TextDiffuser.
> >
> > The GlyphControl dataset, LAION-Glyph-10M, was filtered from LAION-2B-en, and it is likely to have a significant overlap with the text-containing images in LAION-400M. Additionally, in our ablation experiments (refer to Table 2 in Supplementary Material for the latest results), the setup of Exp. 2 is very similar to GlyphControl. It involves disabling editing, text embedding, and perceptual loss while retaining only the glyph image that renders tests based on their positions, along with the corresponding text position image. Under the exact same training dataset, our method (Exp. 8) demonstrates a significant advantage.
> >
> > In addition, we randomly sampled some text lines to assess the accuracy of our OCR annotations. The error rate for the English portion was 2.4%, while for the Chinese portion was significantly higher at 11.8%. This difference may be attributed to the inherent complexity of Chinese character recognition, as well as the performance of PPOCR_v3.
> >
> > In summary, during model training, we only utilized 1.3 million out of 10 million English data and incorporated over 1.6 million non-English data with noise. While our English metrics have a clear advantage over competing methods, this advantage is certainly not derived from the benefits of the data.
> >
> >
> > [1] https://modelscope.cn/models/damo/cv_convnextTiny_ocr-recognition-general_damo/summary

---

> > ### Comment · Reviewer_TuoG · 2023-11-21
> >
> > Another question is about the caption quality. Two recent studies [[1]](https://arxiv.org/pdf/2310.16656.pdf), [[2]](https://cdn.openai.com/papers/dall-e-3.pdf?ref=louisbouchard.ai) shows that improved caption in training diffusion models can significantly improve the image generation quality, and their key message is that the captions should be long and detailed. But you said that the average BLIP caption is only about 8.3 words. Can you explain whether the BLIP captions benefit your model's performance? Can you provide more caption examples of the dataset? Many of the captions for qualitative examples in the main paper start with 'a photo of' or 'a sign of'; is this common in the dataset? There is a section in the appendix describing that the BLIP caption is concatenated with the target words. Does the 8.3-word average length count this part?

---

> > > ### Author Response · Authors · 2023-11-22
> > > **Reply to Reviewer TuoG**
> > >
> > > Dear Reviewer,
> > >
> > > We greatly appreciate your professionalism in this field, and your feedback has helped us to understand your concerns better. We are more than willing to provide as comprehensive a response as possible.
> > >
> > > For the first question, it is indeed the case that the OCR model/dataset primarily provide rectangle bounding boxes as annotations. However, during inference, we may receive input in both the form of rectangle bounding boxes (e.g., during evaluation) and irregular regions (e.g., when users use a brush tool to input a mask). To bridge this gap, we applied data augmentation to the position masks during training to simulate user input. Specifically, each region has a certain probability of being eroded or dilated, using random intensities within a reasonable range, while ensuring that they do not overlap. Then get the polygon of the region using cv2.approxPolyDP( ). As for curved regions, although the OCR model cannot directly output polygon masks, it can output a set of curved rectangle bounding boxes. In these positions, there is a curved text in the training image, and our model can easily learn this correspondence. To provide a clearer understanding, we have included an example in Fig. 8 in Supplemental Material.
> > >
> > > For the second question, long and detailed captions have been proven to have a significant impact on image generation quality, for example, one can produce results comparable to or surpassing those of commercial models by prompt engineering just on SD1.5. The BLIP-2 captions we used are indeed short and lacking in detail, but they are reasonably accurate. We have attempted to use open-source multimodal large models to generate long and detailed captions, but their accuracy is not good enough. Assessing the impact of BLIP-2 on our model's performance is undoubtedly a complex and time-consuming task, and we would like to approach this from a simpler perspective. As described in Appendix A.4, we used positive and negative prompts when generating evaluation images, as ControlNet did(rest assured, the configurations for all methods used in the comparative evaluation are exactly the same). These additional prompts undoubtedly enhance the quality of our captions. Therefore, we removed them all, and re-evaluated our model, as shown below:
> > >  P&N Prompts |Sen. Acc↑(EN) | NED↑(EN) | Sen. Acc↑(CH) | NED↑(CH)
> > > :---:|:---:|:---:|:---:|:---:
> > > w/|0.6509|0.8541|0.6646|0.8274
> > > w/o|0.5897|0.8291|0.6192|0.8108
> > >
> > > We found that the decrease in caption quality indeed led to a certain decrease in performance. In other words, higher-quality captions will undoubtedly further enhance our text generation metrics. However, obviously, this cannot be attributed to our method itself. To provide more caption examples, we have uploaded all the images and JSON annotation files of AnyText-benchmark here: [Google Drive](https://drive.google.com/file/d/1mk6fGTYpee_X7HpwoByauSkJUMbZX0lG/view?usp=sharing). As you asked whether the average length includes concatenated captions, the answer is no, as we stated in our reply to Q8: "...is ~8.26 words per image, **excluding the additional description** of the text to be rendered".
> > >
> > > Hope we have provided a clear explanation of your concerns and look forward to your response. Thank you!

---

> > > > ### Comment · Reviewer_TuoG · 2023-11-22
> > > >
> > > > Thank you for your detailed reply; it resolves most of my questions. I have increased the rating.

---

> ### Comment · Reviewer_TuoG · 2023-11-21
>
> Thanks for the clarification, but I'm still confused about the tight position mask annotation. As far as I understand, not all of the OCR datasets contain polygon mask annotation for the scene texts. For example, MTWI only contains rectangle bounding box annotations. Also, the open-sourced [PPOCRv3](https://github.com/PaddlePaddle/PaddleOCR/blob/release/2.5/doc/doc_en/models_list_en.md#1-text-detection-model) model only predicts rectangle bounding boxes instead of polygon masks. My major concern is how you collected these tight position masks for Anyword-3M. If such tight position mask annotation is very few in the dataset, how can the model generate texts in irregular/curved regions?

---

### Author Response · Authors · 2023-11-23
**General Response to Reviewers**

Dear Reviewers,

We sincerely thank you for the time and effort in reviewing our manuscript!

Generating images with accurate text has been a long-standing issue for diffusion models, especially for non-Latin languages. We are greatly encouraged by your acknowledgment of the **novelty** of our proposed multilingual text generation method, using terms such as *new*(LoJV), *notable*(5GTp, xCSM), and *not achieved in previous works*(TuoG), and also its **effectiveness** with *harmonize*(5GTp), *strategic*(xCSM) and *effective*(TuoG). Additionally, our upcoming **open-sourced dataset** was also mentioned as *new*(LoJV),*useful*(TuoG),and *commendable*(xCSM).

Meanwhile, we have received many insightful and valuable suggestions from you, which have been instrumental in improving our work. We have carefully revised our manuscript, as highlighted in red, and summarized as follows:
- We have improved some details in Sec.3, such as clearly stating that the operation on $z_a$ and $z_t$ involves addition, and clarifying that the feature used in the Text embedding module is from the last fully connected layer of the OCR model. Additionally, we added the linear $\xi$ in Fig.2 and indicated that it is trainable.
- Given that we used the same OCR model, PP-OCRv3, for training and evaluation, this could raise fairness issues (as indeed it did). Therefore, we employed another open-sourced OCR model, DuGuangOCR[1], for the evaluation of Sen.ACC and NED metrics, and updated the values in the quantitative comparison (Table 2) and ablation experiments (Table 3).
- The Text Embedding module has been confirmed to be the most effective module, but it is uncertain how much an OCR model contributes to this. Therefore, we conducted two additional ablation experiments, replacing the PP-OCRv3 with the CLIP vision model (vit) and a stacked convolutional module (conv), to compare the impact of different text feature encoders on performance, as listed in Table 3.
- Even though our method appears to have added many complex components, we have confirmed that the increase in parameter size and computational overhead is almost negligible compared to the vanilla ControlNet, as detailed in Appendix A.2.

All additional concerns and remarks were addressed in individual replies to the reviewers.

Thank you!

[1] https://modelscope.cn/models/damo/cv_convnextTiny_ocr-recognition-general_damo/summary

---

### Meta-Review · Area_Chair_EgXo · 2023-12-14

**Metareview:**

The reviewers are largely impressed by the paper's contributions, particularly its novel approach to improving text portrayal within generated images and its introduction of the AnyWord-3M dataset. While some concerns exist regarding model complexity, data details, and ablation study clarity, I still would like to recommend accept.

The adapter-based module is compatible with existing pre-trained diffusion models and works across multiple languages, addressing a significant gap in previous works. This large-scale dataset with multilingual OCR annotations provides a valuable resource for future text-image generation research.

**Justification For Why Not Higher Score:**

The multi-stage training regimen and numerous hyperparameters may pose challenges for re-implementation. Also, the innovation is not ground-breaking level for an oral presentation.

**Justification For Why Not Lower Score:**

The paper's innovative approach to text-image generation, its state-of-the-art performance on the proposed benchmark, and the valuable AnyWord-3M dataset make it a significant contribution to the field.

---

### Decision · Program_Chairs · 2024-01-16

Accept (spotlight)